# Examining the Use of Antidepressants for Adolescents with Depression/Anxiety Who Regularly Use Cannabis: A Narrative Review

**DOI:** 10.3390/ijerph19010523

**Published:** 2022-01-04

**Authors:** Danielle Hen-Shoval, Aron Weller, Abraham Weizman, Gal Shoval

**Affiliations:** 1Psychology Department, Bar-Ilan University, Ramat Gan 5290002, Israel; danielihen@gmail.com (D.H.-S.); aron.weller@biu.ac.il (A.W.); 2Gonda Brain Research Center, Bar-Ilan University, Ramat Gan 5290002, Israel; 3Geha Mental Health Center, Petah Tiqva 4910002, Israel; aweizman@clalit.org.il; 4Sackler Faculty of Medicine, Tel Aviv University, Tel Aviv 6997801, Israel; 5Princeton Neuroscience Institute, Princeton University, Princeton, NJ 08540, USA

**Keywords:** antidepressants, depression, anxiety, cannabis, tetrahydrocannabinol (THC), cannabidiol (CBD)

## Abstract

Depression and anxiety disorders are two of the most common and growing mental health concerns in adolescents. Consequently, antidepressant medication (AD) use has increased widely during the last decades. Several classes of antidepressants are used mainly to treat depression, anxiety, and obsessive-compulsive disorders by targeting relevant brain neurochemical pathways. Almost all randomized clinical trials of antidepressants examined patients with no concomitant medications or drugs. This does not address the expected course of therapy and outcome in cannabis users. Cannabis is the most commonly used illicit substance globally. Substantial changes in its regulation are recently taking place. Many countries and US states are becoming more permissive towards its medical and recreational use. The psychological and physiological effects of cannabis (mainly of its major components, tetrahydrocannabinol (THC) and cannabidiol (CBD)) have been extensively characterized. Cannabis use can be a risk factor for depressive and anxiety symptoms, but some constituents or mixtures may have antidepressant and/or anxiolytic potential. The aim of this literature review is to explore whether simultaneous use of AD and cannabis in adolescence can affect AD treatment outcomes. Based on the current literature, it is reasonable to assume that antidepressants are less effective for adolescents with depression/anxiety who frequently use cannabis. The mechanisms of action of antidepressants and cannabis point to several similarities and conjunctions that merit future investigation regarding the potential effectiveness of antidepressants among adolescents who consume cannabis regularly.

## 1. Introduction

Major depression disorder (MDD) is the prominent cause of disability-adjusted life years lost in 10–19-year-olds, with a global prevalence of 4–25% [1,2,3]. Since COVID-19 was declared an international public health emergency, youth around the world have experienced dramatic disruptions to their everyday lives, which has increased this prevalence and the associated disability [4]. This matter has special importance because it is considered a ‘gateway’ disorder, increasing the likelihood of adult depression and other psychiatric disorders later in life, with associated social, medical, and economic sequelae [5,6,7,8].

Whereas MDD manifests as an episodic but often recurrent illness with a mean duration of 16 weeks, often comorbid with other mental disorders, particularly anxiety [9], the clinical presentation of MDD in adolescents is quite different from the adult one. In fact, it is characterized by heterogeneous and changing symptoms, sometime hidden from somatic complaints and complicated by the high comorbidity rates with anxiety disorders, substance abuse, disruptive behavior disorders, personality disorders, and medical illnesses [10,11,12,13]. A main concern for this is the risk for suicide. In adolescents as well as in adults, although suicide is a complex phenomenon which has many causes, 85% to 95% of those dying by suicide have a psychiatric illness (particularly MDD) [14], and there is an over-representation of individuals with substance-use disorders [15,16,17,18,19]. Furthermore, there are significant psychosocial and educational consequences if such an episode remains undetected [20]. Therefore, in these individuals it is difficult to recognize depressive symptoms and make a correct diagnosis, as well as to establish an adequate therapeutic strategy [21].

Evidence-based treatment for depression and anxiety consists of psychological therapies, e.g., cognitive behavioral therapy, and antidepressant medication (AD) [22]. However, because of insufficient resources, antidepressants are used more frequently than psychological interventions [23] In particular, the rate of its prescription among adolescents has increased over time [24]. Thus, AD has long been the basis of medical treatment for depression and a suitable replacement for benzodiazepines in long-term treatment of anxiety disorders [22]. Although clinical guidelines recommend AD for minimum 6 months following response in MDD and anxiety disorders, medication adherence rates in adolescents are poor [25]. There has been continuous debate about AD efficacy [23,26,27,28], though they remain an effective, yet not fully satisfactory, treatment option for many patients with depression and anxiety [29].

Less is known about AD interaction with cannabis. For instance, whereas ADs are associated with reduction of suicidal behavior [30], recreational cannabis use is often associated with increased suicidality, at least in some populations [31]. It is critical to recognize the joint effect of ADs when consumed together with medical or non-medical cannabis.

Cannabis, commonly known as marijuana, contains Δ-9-tetrahydrocannabinol (THC) and cannabidiol (CBD), the main constituents, and a plethora of phytocannabinoids alongside a vast array of flavonoids and terpenes [32]. It is the most widely used illicit substance worldwide [33]. Over the past 50 years, not only has cannabis been increasingly adopted by adolescents and young adults for recreational use, mainly in social settings to increase sociability and produce euphoric and tranquilizing effects [34], but the age of use initiation is shifting back with younger children and adolescents reporting daily cannabis use [35]. Although cannabis remains illegal in most countries, there have been significant changes in its use as a therapeutic medicine [36,37,38]. Accumulating evidence suggests that some cannabinoids, particularly CBD, may be an effective and safe anxiolytic and potentially also antidepressant agents [32,39,40,41,42,43,44].

A core problem is that use amongst depressed teens has increased more rapidly over the past 15 years compared to their peers [45], yet, almost all clinical studies of ADs examined ‘clean’ patients; with no concomitant use of psychoactive substances. This does not represent the course of therapy in cannabis users.

Hence, the aim of the current literature review is to obtain a deeper understanding regarding the possible effects of ADs efficiency for adolescents who regularly use cannabis.

We hypothesized that due to some similarities between the mechanisms of action of antidepressants and cannabis, the outcomes of AD treatment are limited.

## 2. Method

### Search Strategy and Paper Selection

A comprehensive, two-stage, literature search was performed for all papers published up to 30 October 2021, using PubMed, SCOPUS, and Google Scholar electronic databases. In the first stage, we used the search terms: “Depression”, or “Antidepressants” or “Anxiety”, together with “Cannabis”, or “Cannabidiol”, or “THC”, or “Cannabinoids”. We included all original research papers, review articles, non-research letters, communications, commentaries meeting search criteria, but excluded case reports, small case series, and non-English language articles. One independent author (D.H.) screened the titles and abstracts for potential articles. In the second stage, papers with these combined terms on “Adolescents” or “Adolescence” were prioritized. The full texts of remaining articles were evaluated by applying the inclusion and exclusion criteria. Even though this is a narrative review, we have attempted to present our search strategy including approximate numbers of papers at each stage in a PRISMA type flowchart (see Figure 1).

## 3. The Efficacy of Antidepressants: The Importance of Adherence

ADs are the frontline intervention for treatment of anxiety and depressive disorders and obsessive-compulsive disorder (OCD), in both adults and youngsters, with and without co-morbidity [46,47,48,49,50,51]. Clinicians have a wide choice of drugs [23]. While the prevalence of anxiety and MDD remained stable over the past decades [52], use of antidepressants is increasing worldwide [53,54]. This began in the mid-1950s with the introduction of tricyclic antidepressants (TCAs) (e.g., imipramine) and monoamine oxidase inhibitors (MAOIs) (e.g., iproniazid). Both classes of agents produced a dramatic improvement in depression symptoms but had poor tolerability and safety profiles [55]. Subsequently, along with recognizing the importance of the serotonergic system in pathophysiology and treatment of depression [56], second-generation medications were developed: selective serotonin reuptake inhibitors (SSRIs) (e.g., fluoxetine), selective norepinephrine (e.g., reboxetine), dual serotonin and norepinephrine reuptake inhibitors (SNRIs) (e.g., venlafaxine), norepinephrine/dopamine reuptake inhibitor (NDRIs) (e.g., bupropion), and noradrenergic and specific serotonergic antidepressants (NaSSAs) (e.g., mirtazapine) (Table 1).

In spite of the contemporary excessive use of these second-generation drugs, there is a long-lasting debate and concern about their efficacy and effectiveness. Treatment often takes several weeks or months to reach their full therapeutic effects [56], and 30–50% of the patients do not respond to treatment with ADs [57]. In fact, no AD or class of ADs offers a faster onset of action [58] and no single AD treatment is uniformly effective [59,60]. Although international practice guidelines recommend that pharmacotherapy for adolescents should be initiated with fluoxetine, with sertraline or citalopram used in case of non-response to fluoxetine [61], after unsuccessful treatment for depression with an SSRI, it is unclear whether switching to a particular antidepressant is more effective than switching to another [62].

This heterogeneity in the effectiveness in AD treatment is also manifested differently by gender as some studies indicated that men respond better to TCAs than to SSRIs whereas women tend to show the opposite pattern of response [63]. Numerous studies attempt to explain these discrepancies [64,65,66] yet the reason for this remains unclear [67]. Surprisingly, little is known about these differences among adolescents. Another important dimension is period of life, as effectiveness of AD treatment varies across ages [68,69,70,71]. For example, young adults had a lower response to noradrenergic antidepressants than they did to serotonergic antidepressants, whereas there was no differential response in the older age group [70]. The reason for these differences is poorly understood [68].

The primary goal of AD treatment maintenance is to prevent a subsequent episode of depression (recurrence), anxiety, or OCD symptoms and development of chronicity [72,73,74]. However, this goal is not achieved in many cases. While a remitted patient with depression or anxiety is symptom-free [74,75], the probability of achieving and sustaining symptomatic remission for adolescence with MDD with first-line pharmacotherapy is as low as approximately 30–40% [76]. This probability is similar for adolescents with anxiety. For example, in patients with generalized anxiety disorder (GAD), sertraline has demonstrated remission rates that are 34% to 46% higher than placebo [77]. Regarding OCD, studies report 25% to 47% symptomatic remission rates [78]. In fact, long-term AD use may increase, in some cases, biochemical vulnerability to develop depressive episodes and worsen long-term outcome and symptomatic expression of MDD, decreasing likelihood of subsequent response to pharmacologic treatment and duration of symptom-free period [59].

ADs have very broad mechanisms of action which lead to a variety of adverse effects related to their potent activity on cholinergic, adrenergic, and histaminergic receptors [55,79,80]. This is reflected in numerous side effects such as sexual dysfunction [81,82], headaches [83], somnolence [83] and weight gain [81,83,84]. Additionally, these side effects may consequently lead to decreased adherence to AD [85], which is another key issue regarding currently available AD compounds. According to a growing body of research, the effectiveness of AD treatment is highly influenced by adherence rates [86,87,88], since using AD continuously is a key to successful treatment outcomes [89] and reduced likelihood of relapse or recurrence of depression [90]. For AD to be efficacious, patients must remain adherent until symptoms remit, up to 12 weeks [91,92]. To decrease the risk of relapse, the American Academy of Child and Adolescent Psychiatry (AACAP) Practice Parameters recommend continuation of treatment for 6–12 months “for all patients who have responded to the acute treatment” [93]. Moreover, adherence to AD is associated with lower risk of premature mortality [94] in various AD-using populations [51,95,96,97]. However, low adherence rate is commonly reported [95,96,97,98,99,100,101]; about 60% of patients discontinue antidepressants within 3 months [102,103], with both genders following a similar pattern of AD adherence across age decades [104]. It is still unclear whether the limited efficacy of AD is due to low adherence rates or maybe the causal relationship is vice versa. Such putative bi-directional influence may lead to a positive feedback loop or “a vicious cycle” with mounting negative effect.

In the last decade, many studies suggest that cannabinoids be explored as potential novel ADs.

## 4. Cannabinoids for Treatment of Depression and Anxiety: Changing Perceptions throughout the Years

Over the last 25 years the endocannabinoid system (ECS) has emerged as an important neuromodulatory system [105], which includes ligands, enzymes, and endogenous cannabinoid receptors, widespread throughout the brain and parts of the body [106]. Although this system is activated during consumption of illegal drugs containing exocannabinoids such as marijuana and synthetic cannabinoids [107], research suggests it is important for regulation of many basic physiological functions such as cognition, learning, memory, perception, sleep, pain, appetite, motor control, and regulation of cardiovascular and immune responses [108,109,110,111,112,113,114].

A fundamental element in the discovery of the ECS is *Cannabis sativa* L., which has a long history as a medicinal plant [115]. Since the discovery of its main psychomimetic constituent, Δ9-Tetrahydrocannabinol (Δ9-THC), about half a century ago [116], studies have shown that it produces many additional compounds, including various phytocannabinoids [117]. The concentration of these compounds depends on tissue type, age, variety, growth conditions (nutrition, humidity, light level), harvest time, and storage conditions [118], leading to a wide range of pharmaceutical effects [119]. To date, out of the ±150 cannabinoids that have been identified, the most studied and most active are THC and CBD [32]. CBD, unlike THC, is devoid of psychotomimetic effects [120]. In general, THC and CBD seem to have opposite effects [121,122,123,124,125] (Figure 2).

It is important to distinguish between the multi-purpose applications of cannabis in the context of medicinal and social purposes. It is considered a controversial plant due to its recreational use [126], highlighted by the ‘social high’ induced by marijuana (usually predominantly comprised of THC [127]) [128,129]. Since the recreational use of cannabis was first reported [130], it has spread globally, first to high-income countries, then to low- and middle-income countries [131,132]. Likewise, the National Epidemiologic Survey on Alcohol and Related Conditions (NESARC) longitudinal epidemiological study found that, between 2001–2002 and a decade later (2012–2013), marijuana use and Diagnostic and Statistical Manual of Mental Disorders Diagnostic and Statistical Manual of Mental Disorders (DSM)-IV marijuana disorder prevalence doubled [133].

Among the most prominent concerns regarding cannabis use is the connection to mental disorders [134]. Numerous studies support ECS involvement in modulation of the hypothalamic–pituitary–adrenal (HPA) axis; regulation of mood and reward, anxiety, and extinction of fear learning [135,136,137,138,139]. This is expressed both in psychotic and non- psychotic disorders when a large intake of cannabis appears to trigger acute psychotic episodes and worsen outcomes in existing psychosis [140,141,142,143]. In parallel, a growing body of evidence supports an association between cannabis use and depression and anxiety [144,145,146], including in young adults. Cannabis consumption during adolescence is associated with increased risk of developing MDD and suicidality, especially suicidal ideation in young adulthood [147] which is manifested differently across gender [148]. Moreover, it places them at greater risk for maintaining higher levels of anxiety over time [149].

On the other hand, cannabis psychoactive preparations have been used for over 4000 years for medical purposes [150] and a great portion of its medical efficacy is attributed to CBD’s neuroprotection properties [151] which include antioxidant and anti-inflammatory activities [152,153].

From a psychiatric perspective, CBD is a novel promising therapeutic agent. It attenuates the psychotic-like effects of cannabis over time in recreational users [154] and it was repeatedly shown to induce anxiolytic activity in preclinical and clinical studies [41,155,156,157] in addition to its anti-compulsive effects [158,159,160,161]. Furthermore, it exhibits anti-depressive-like abilities in several animal models [42,43,44], yet, to the best of our knowledge, there is no published controlled clinical study that has investigated whether CBD can decrease depressive symptoms in patients. A laboratory study reported that CBD attenuated the transient “amotivational” effects of THC-based cannabis [162] and a small-scale clinical trial focusing on different outcomes reported that depression was an intervening variable in the effects of Sativex (which has a high ratio of CBD:THC) on the main topic of interest [163]. A potential treatment for borderline personality disorder (BPD), based on ratio of high level of CBD to low level of THC has been suggested [32] and CBD-based compounds have been found to be potent in the relief of anxious and depressive symptoms [164,165,166,167].

Unfortunately, medical use of cannabis has been shown to be significantly associated with non-medical use of cannabis [168]. This overlap presents new difficulties when psychiatric patients regularly use cannabis.

## 5. Antidepressant Treatment Combined with Cannabis Use: A Gap in Knowledge

The knowledge about interactions between cannabis use and long-term AD treatment is vague. HPA axis dysregulation plays a role in vulnerability to stress-related disorders, such as anxiety and depression. AD agents normalize its hyperactivity [169,170,171,172]. The ECS can also regulate the HPA axis activity [173,174,175,176]. Accordingly, deficits in ECS signaling may result in depressive and anxiogenic behavioral responses, while pharmacological augmentation of endocannabinoid signaling can produce both anti-depressive and anxiolytic behavioral responses [177].

This duplicity may possibly be attributed, at least in part, to the bidirectional effects of cannabinoids on anxiety, with low doses having anxiolytic and high doses anxiogenic effects, as well as to the individual’s history and the environmental context [178]. Thus, CBD’s effect on depressive-like behavior in mice has an inverted U-shaped dose-response curve [44], whereas the association between MDD and cannabis use in humans is more complex. Unfortunately, many longitudinal studies exploring the association between cannabis use and MDD employed unclear categories for defining frequency of cannabis use or did not record the frequency of cannabis use [179], while greater exposure to cannabis is expected to lead to a greater incidence of MDD at follow-up [180].

The meaning of this is that combined cannabis and AD use interaction may be dose-dependent. In general, low doses of cannabis are stimulatory as they were found to be anxiolytic, whereas high doses are inhibitory and anxiogenic [178]. This means that any effect of this interaction should be explored carefully, taking into account dose-use patterns of cannabis users. Due to the pivotal role of the ECS in the regulation of emotional states, it is most likely that a patient who uses AD will be affected differently by this combined use of cannabis whether he/she consumes stimulating low doses or inhibitory high doses. In addition, one must take into consideration the relatively unpredictable nature of the response of humans to cannabis consumption, which is derived from multiple factors.

One possible explanation for these biphasic effects of cannabinoids is that distinct receptors with differential sensitivity to cannabinoids are implicated in their inhibitory/anxiogenic and stimulatory/anxiolytic effects [178]. Precise short-term and long-term dose-related effects of cannabinoids in humans remain to be studied carefully [181].

Another aspect that requires examinations is the possible drug–drug interactions resulting from the pharmaco-metabolic processes of ADs when combined with cannabinoids. This issue has not been explored fully; yet, a few similarities between these compounds’ mechanisms of action can point to a competitive effect. Although there are limited studies of direct SSRI–cannabinoid interactions, accumulating data suggest the potential for interactions [182].

Longstanding evidence suggests that changes in activity of 5HT1A serotonergic inhibitory auto-receptors, mediating cortisol, and ACTH secretion, as well as the regulation of serotonergic neuronal firing [183], mediate symptom improvement during antidepressant therapy [184]. Specifically, SSRIs modulate the serotonergic (5HT) system and the HPA axis by affecting sensitivity of 5HT1A auto-receptors [184,185,186], an effect thought to be a necessary prerequisite for clinical response [187,188,189]. Similarly, anxiolytic effects produced by cannabinoids are modulated by 5-HT1A receptors [190,191]. This was also confirmed for CBD, as shown in many animal studies [44,192,193,194].

Additionally, changes in neuronal plasticity and BDNF signaling have been implicated in the etiology of depression and in AD drug action [193]. Specifically, ADs increase the synthesis of BDNF [195,196,197,198,199]. Similarly, cannabinoids also affect BNDF levels. In preclinical studies, THC has been shown to alter BDNF expression [200,201,202,203] and CBD increases BDNF signaling in models of neurodegeneration [204,205,206]. However, this effect may differ by the individual’s cannabis consumption habits; intravenous administration of THC increased serum BDNF levels in healthy controls but not in occasional cannabis users and that the latter have lower basal BDNF levels [207].

Furthermore, gut microbiota have been implicated in regulation of pathophysiology of several mental disorders, including anxiety and depression [208,209,210]. Drugs belonging to new classes of ADs have antimicrobial effects [211]. For instance, the gut microbiota–brain axis at least partially mediates the antidepressant actions of (R)-ketamine [212] and vagus nerve dependent gut-brain signaling contributes to the effects of oral SSRI [213]. Interestingly, cannabinoids and the ECS are involved in regulating the gut microbiome [214,215,216,217], with some recent evidence supporting the effects of cannabis consumption on the microbiota–gut–brain axis [218,219].

One more consideration in the overlap between AD treatment and cannabis use is in their rich cross-talk with the immune system. While accumulating evidence indicates immunomodulatory and anti-inflammatory effects of cannabinoids [220,221,222,223,224], the connection of stress, depression, and anxiety with the immune system is well established [225,226,227,228,229,230,231,232]. The meaning of this interaction is that depressed and anxious patients may already have immune disruption [225,226,228,229,230,231,232], and that use of AD normalizes it [232]. Due to the fact that the precise mechanisms whereby antidepressants cause these changes are uncertain, it is unclear whether additional cannabis use, which affects the immune system as well [220,221,222,223,224], may affect AD treatment’s efficiency.

Another important issue to be considered is that cannabis consumption while on AD treatment can potentially cause the patient to use AD improperly. Cannabis is generally used to elevate mood [233]. It enables and increases the subjective sense of well-being [234]. Furthermore, this combination of cannabis and AD can affect the stability of the patients’ mental states, making it hard for them to distinguish between the physical and psychological effects derived from the cannabis use and their AD treatment. A situation like this may lead to mistaken conclusions. Thus, it is reasonable to believe that cannabis use may affect adherence to AD: patients might use their AD medication less than prescribed or even chose to use more cannabis or other drugs instead of the prescribed AD. Indeed, cannabis use is strongly related to use of other drugs [235] which can lead to a greater complication.

In summary, the examination of AD and cannabis mechanisms of action points at several similarities between these two, which can lead to modulatory effects of cannabis on AD’s effectiveness. Still, this issue merits in-depth investigation before drawing conclusions. At present, it is unclear whether ADs are effective for depressive/anxious patients who use cannabis frequently.

## 6. Summary: What Is Known and What Needs to Be Studied about Antidepressants and Cannabinoids?

The rate of cannabis use while on AD medication is an important issue that became more relevant in the last decades due to the sharp increases in both AD prescription [53] and cannabis use [236]. This is accompanied with a worldwide trend toward liberalizing cannabis policy and commercializing its sale [129]. The upsetting reality is that cannabis use was more than twice as common and increased more rapidly from 2004 to 2016 among youth with depression compared to youth without depression [45].

This dearth of knowledge merits future basic and clinical studies on this important issue. Whereas a thoughtful examination of this is vital to clarify the possible effects, there are a few significant limitations regarding future research. First, an accurate measurement of cannabis use is challenging. There is a connection between mixing different types of cannabis; recreational use, which is considered as non-medical, i.e., in social settings [34], and the use of cannabinoids in various medical conditions [237,238,239]. However, these two uses are significantly associated [168]. Furthermore, it is especially important to consider the level of cannabis use. Some studies grouped cannabis abuse and dependence into ‘use disorders’ [240], in line with DSM-5 guidelines [241], which correlate with diagnosis of depression/anxiety and AD treatment, while other studies have used different controversial levels of use for ‘heavy’ and ‘light’ cannabinoid use [242,243,244]. In addition, most research has focused on THC, CBD, and their dissimilarities regarding depression and anxiety. However, ‘cannabis’ is not a single compound product [32]. Although it is known that both non-medical and medical marijuana use could contribute differentially to clinical outcomes and potentially lead to barriers to mental health care in this population, it is problematic to achieve a valid measurement by estimating the influence of each compound separately. Considering that cannabis inflorescences accumulate hundreds of milligrams of terpenes alongside cannabinoids [32], the ratio of THC:CBD:other chemicals/terpenes in cannabis is frequently unknown, making it difficult to assess the exact ratio of chemicals and their effects on AD treatment. Consequently, it is crucial to specify unambiguous definitions for the level of use and ratio of compounds in the cannabis before exploring its effect on AD treatment.

Secondly, much of the research on CBD and depression/anxiety is preclinical. To the best of our knowledge, there is no published controlled clinical study that has explored the effect of CBD on depression, along with little evidence regarding its effect on anxiety in humans. Perhaps a significant amount of human data may provide new insights for a better understanding of the innovative combinations of it with AD.

## 7. Conclusions

There is relatively scarce information about the pharmacological interactions between these two groups of drugs and the clinical efficacy of AD when prescribed to a cannabis-using patient is unknown. Hence, the answer as to whether ADs are effective for patients with depression/anxiety who use cannabis remains open.

Based on current literature and current patterns of AD use in adolescence, we assume that cannabis use while combined with ADs can affect ADs’ pharmaco-metabolic processes and lead to adverse long-term effects. Given the similarities between cannabis compounds and AD mechanisms of action, there are some expected effects which may likely diminish the positive outcome and intensify the side effects of AD treatment.

To our knowledge, only one study has examined this possible interaction of marijuana and AD use among adolescents [182]. Nevertheless, this matter warrants intensive research before a definitive conclusion can be drawn.

## Figures and Tables

**Figure 1 ijerph-19-00523-f001:**
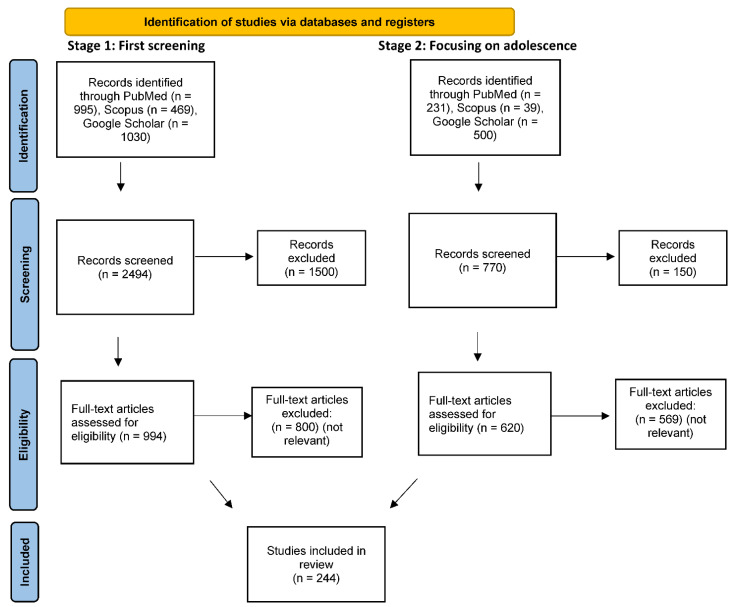
PRISMA 2020 flow diagram for retrospectively estimated numbers of papers at each stage of screening.

**Figure 2 ijerph-19-00523-f002:**
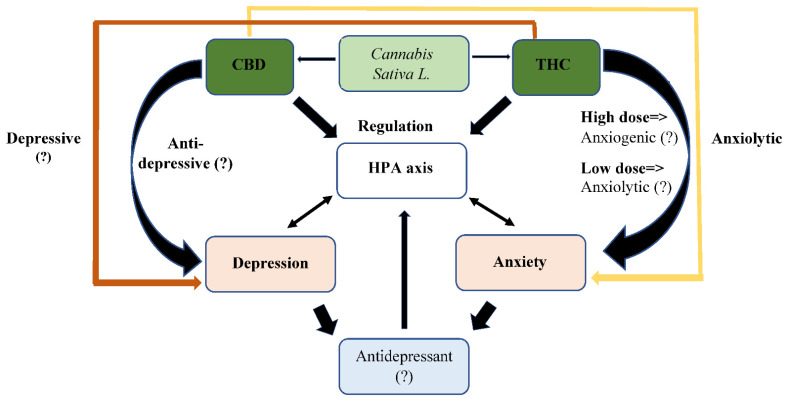
Putative effect of cannabinoids on depression and anxiety. Some of the data presented above is speculative and needs further clarification in future study. In addition, the putative role of additional cannabinoids contained in cannabis has not been well studied and is not represented in this figure.

**Table 1 ijerph-19-00523-t001:** Antidepressant classes in clinical use among patients diagnosed with MDD, anxiety, and OCD.

Drug Name	Active Principle	Main Targets	Mechanisms of Action	Main Side Effects
Tricyclic antidepressants (TCAs)	The chemical structure of a TCA consists of a three-ringed structure with an attached secondary or tertiary amine	Serotonin, norepinephrine, and acetyl choline	Act on approximately five different neurotransmitter pathways to achieve their effects: block the reuptake of Serotonin and Norepinephrine in presynaptic terminals which leads to increased concentration of these neurotransmitters in the synaptic cleft. act as competitive antagonists on post-synaptic alpha cholinergic (alpha1 and alpha2), muscarinic, and histaminergic receptors (H1).	Constipation, dizziness, blurred vision, confusion, urinary retention, and tachycardia
Monoamine oxidase inhibitors (MAOIs)	Blocking monoamine oxidase	Norepinephrine and serotonin	Breaks down different types of neurotransmitters from the brain: norepinephrine, serotonin, dopamine, and tyramine. MAOIs inhibit the breakdown of these neurotransmitters thus, increasing their levels.	Dry mouth, nausea, diarrhea, constipation, drowsiness, insomnia, dizziness, and/or lightheadedness
Selective Serotonin Reuptake inhibitors (SSRIs)	Inhibit the reuptake of serotonin	Serotonin	Block the reuptake of serotonin into the presynaptic nerve terminal via the serotonin uptake site, thus increasing the synaptic concentration of serotonin.	Flatulence, somnolence, memory impairment, decreased concentration, yawning, fatigue, dry mouth, weight gain, light headedness, adverse sexual effects, and sweating
Selective Norepinephrine	Inhibit reuptake of norepinephrine	Norepinephrine	Block the reuptake of norepinephrine into the presynaptic nerve terminal via somatodendritic 2a-adrenoceptors, thus increasing the synaptic concentration of norepinephrine	Dry mouth, constipation, insomnia, increased sweating, tachycardia, vertigo, urinary hesitancy and/or retention, and impotence
Dual Serotonin and Norepinephrine Reuptake Inhibitors (SNRIs)	Inhibit the uptake of serotonin and norepinephrine	Norepinephrine and serotonin	Bind to serotonin and norepinephrine transporters to selectively inhibit the reuptake of these neurotransmitters from the synaptic cleft	Nausea, hypertension, somnolence, dizziness, and dry mouth
Norepinephrine/Dopamine Reuptake Inhibitor (NDRIs)	Inhibit the uptake of dopamine and norepinephrine	Norepinephrine and dopamine	Block the reuptake of norepinephrine and dopamine into the presynaptic nerve terminal thus increasing the synaptic concentration of norepinephrine and dopamine	Fatigue, sleepiness, and somnolence
Noradrenergic and Specific Serotonergic antidepressants (NaSSAs)	Enhance serotonergic and noradrenergic neurotransmission	Norepinephrine and serotonin	Potent antagonism of central α2-adrenergic autoreceptors and heteroreceptors and antagonism of both 5-HT2 and 5-HT, receptors with low affinity for muscarinic, cholinergic, and dopaminergic receptors	Somnolence, increased appetite, weight gain, dry mouth, constipation, and dizziness

## Data Availability

Not applicable.

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
