# Peer review of "Examining the Use of Antidepressants for Adolescents with Depression/Anxiety Who Regularly Use Cannabis: A Narrative Review"

_ijerph, 2022, doi:10.3390/ijerph19010523_

Round 1

Reviewer 1 Report

In attachment.

Author Response

Please see inside attached file.

Reviewer 2 Report

Dear colleagues,
I understand that you have done a good narrative review of the literature. 
But in my opinion, this type of work should be accompanied by a detailed report on the search strategy, type of studies analyzed, in short, a good description of the methodology you have followed.
This does not appear in your work, so I think it does not meet the minimum requirements to be published.

Best regards

Author Response

Please see inside the attached file.

Reviewer 3 Report

The authors of the review “Examining the use of antidepressants for adolescents with depression/anxiety who regularly use cannabis” discussed the importance of implement the acquisition of new data on the interaction between antidepressants and cannabis in adolescents.

Overall, the article is well written and gives the reader a clear idea of the modern context in which cannabis and antidepressants may interact in adults and adolescents.

Few points should be addressed to improve the article:

  • For paragraph 2 “The efficacy of antidepressants: The importance of adherence” a table summarizing drug name, active principle, target, mechanism of action and side effects of the drugs mentioned would help the reader tremendously
  • For paragraph 3 “Cannabinoids for treatment of depression and anxiety: Changing perceptions 154 throughout the years” I suggest the authors to include another table or image describing the active principles of Cannabis sativa L., its effects on depression, and supporting references

Minor corrections:

  • Line 61: “it remains” instead of “they remain”
  • Line 124: please add % after 40 (30%-40% or 30-40%)
  • Line 164: For Cannabis sativa the complete name of the species and the italic diction should be used: Cannabis sativa
  • Line 166: after constituent the semicolon should be replaced by a comma
  • Line 181: DSM stands for Diagnostic and Statistical manual of mental disorder? If so, please include it in the text
  • Line 233-234: The authors state “In general, low doses of cannabis are stimulatory whereas high doses are inhibitory”. Please specify what's the main effect of AD-cannabis interaction here. In other words, what this interaction stimulates and what inhibits.
  • Line 258-259: “it was suggested that intra-258 venous administration of THC increased serum…”. It was observed or suggested? Please specify.
  • Line 271-272: The authors should expand this aspect in few sentences. What's known about immune system/ cannabinoids interaction and depression?

Author Response

Please see inside the attached file.

Round 2

Reviewer 2 Report

Dear authors,
I appreciate the inclusion of a paragraph about the search strategy but in my opinion the description of it is very scarce.
Even though it is a narrative review, you should include aspects that are considered important to ensure a correct methodology.
I suggest that you consult the PRISMA Statement.
http://www.prisma-statement.org/Default.aspx

Best regards

Author Response

Thank you for the opportunity to make additional revisions to our manuscript. We are uploading the revised version together with a new figure (a flowchart, as requested).

Briefly, in accordance with the comments by a reviewer and by the editor, we have:

  1. Revised the title to explicitly state that the paper is a narrative review
  2. Added aims and hypothesis in the Abstract and at the end of the Introduction.
  3. Added a detailed flowchart typical of the PRISMA approach (Figure 1) in which each stage of the search strategy is depicted along with the estimated (retrospectively) number of papers examined at each stage. 

We hope that this re-revised version will be acceptable for publication in IJERPH